# STRUCTURED EXPLORATION VIA HIERARCHICAL VARIATIONAL POLICY NETWORKS

## ABSTRACT

Reinforcement learning in environments with large state-action spaces is challenging, as exploration can be highly inefficient. Even if the dynamics are simple, the optimal policy can be combinatorially hard to discover. In this work, we propose a hierarchical approach to structured exploration to improve the sample efficiency of on-policy exploration in large state-action spaces. The key idea is to model a stochastic policy as a hierarchical latent variable model, which can learn low-dimensional structure in the state-action space, and to define exploration by sampling from the low-dimensional latent space. This approach enables lower sample complexity, while preserving policy expressivity. In order to make learning tractable, we derive a joint learning and exploration strategy by combining hierarchical variational inference with actor-critic learning. The benefits of our learning approach are that 1) it is principled, 2) simple to implement, 3) easily scalable to settings with many actions and 4) easily composable with existing deep learning approaches. We demonstrate the effectiveness of our approach on learning a deep centralized multi-agent policy, as multi-agent environments naturally have an exponentially large state-action space. In this setting, the latent hierarchy implements a form of multi-agent coordination during exploration and execution (`MACE`). We demonstrate empirically that `MACE` can more efficiently learn optimal policies in challenging multi-agent games *with a large number ($\sim$ 20) of agents*, compared to conventional baselines. Moreover, we show that our hierarchical structure leads to meaningful agent coordination.

## 1 INTRODUCTION

Reinforcement learning in environments with large state-action spaces is challenging, as exploration can be highly inefficient in high-dimensional spaces. Hence, even if the environment dynamics are simple, the optimal policy can be combinatorially hard to discover. However, for many large-scale environments, the high-dimensional state-action space has (often hidden or implicit) low-dimensional structure which can be exploited.

Many natural examples are in collaborative multi-agent problems, whose state-action space is exponentially large in the number of agents, but have a low-dimensional coordination structure. Consider a simple variant of the Hare-Hunters problem (see Figure 1). In this game, $N = 2$ identical hunters need to capture $M = 2$ identical *static* prey within $T$ time-steps, and exactly $H = 1$ hunter is needed to capture each prey. $T$ is set such that no hunter can capture both preys. There are two equivalent solutions: hunter 1 captures prey 1 and hunter 2 captures prey 2, or vice versa. There are also two suboptimal choices: both hunters choose the same prey.

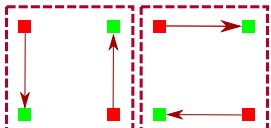

Figure 1: Equivalent solutions in a 2-hunter 2-prey game.

Hence, the hunters must coordinate *over a (large) number of time-steps* to maximize their reward. This implies the solution space has low-dimensional structure that can be used to accelerate training.

In this work, we propose a principled approach to structured exploration to improve sample complexity in large state-action spaces, by learning deep hierarchical policies with a latent structure. As a high-level intuition, consider a tabular multi-agent policy, which maps discrete (joint) states to action probabilities. For $N$ agents with $S$ states and $A$ actions each, this policy has $O((S \cdot A)^N)$ weights. However, the low-dimensional coordination structure can be captured by a factorized, low-rank

matrix, where the factorization can be learned and, for instance, only has $O(NK(S + A))$ weights. Similarly, our approach both 1) learns a low-dimensional factorization of the policy distribution and 2) defines exploration by also sampling from the low-dimensional latent space. For instance, in the multi-agent setting, we can learn a centralized multi-agent policy with a latent structure that encodes coordination between agents and biases exploration towards policies that encode "good" coordination.

The key ideas of our approach are: 1) to utilize a shared stochastic latent variable model that defines the structured exploration policy, and 2) to employ a principled variational method to learn the posterior distribution over the latents jointly with the optimal policy. Our approach has several desirable properties. First we do not incorporate any form of prior domain knowledge, but rather discover the coordination structure purely from empirical experience during learning. Second, our variational learning method enables fully differentiable end-to-end training of the entire policy class. Finally, by utilizing a hierarchical policy class, our approach can easily scale to large action spaces (e.g. a large number of coordinating agents). Our approach can also be seen as a deep hierarchical generalization of Thompson sampling, which is a historically popular way to capture correlations between actions (e.g. in the bandit setting (Agrawal & Goyal, 2012)).

To summarize, our contributions in this work are as follows:

- We introduce a structured probabilistic policy class that uses a hierarchy of stochastic latent variables.
- We propose an efficient and principled algorithm using variational methods to train the policy end-to-end.
- To validate our learning framework, we introduce several synthetic multi-agent environments that explicitly require team coordination, and feature competitive pressures that are characteristic of many coordinated decision problems.
- We empirically verify that our approach improves sample complexity on coordination games with a large number ($N \sim 20$) of agents.
- We show that learned latent structures correlate with meaningful coordination patterns.

## 2 COOPERATIVE MULTI-AGENT REINFORCEMENT LEARNING

We use multi-agent environments to show the efficacy of our approach to structured exploration, as they naturally exhibit exponentially large state-action spaces. In this work we focus on efficiently learning a *centralized policy*: a joint policy model for all agents, in the full-information setting. More generally, multi-agent problems can be generalized along many dimensions, e.g. one can learn decentralized policies in partial-information settings. For an overview, see Busoniu et al. (2008).

In multi-agent RL, agents sequentially interact within an environment defined by the tuple: $\mathcal{E} \equiv (\mathbf{S}, \mathbf{A}, \mathbf{r}, f_P)$. Each agent $i$ starts in an initial state $s_0^i$, and at each time $t$ observes a state $\mathbf{s}_t \in \mathbf{S}$ and executes an action $a_t^i$ chosen by a (stochastic) policy $a_t^i \sim P\left(a_t^i|\mathbf{s}_t\right)$. Each agent then receives a reward $r^i\left(\mathbf{s}_t, \mathbf{a}_t\right)$, and the environment transitions to a new state $\mathbf{s}_{t+1}$ with probability $f_P\left(\mathbf{s}_{t+1}|\mathbf{s}_t, \mathbf{a}_t\right)$. We define the joint state and actions as $\mathbf{s}_t = \{s_t^i \in \mathbf{S}\}$ and $\mathbf{a}_t = \{a_t^i \in \mathbf{A}\}$, where $i \in \mathcal{I}$ indexes the agents. Note that the rewards for each agent $r^i$ can depend on the full joint state and actions.

In this work, we restrict to fully cooperative MDPs that are fully observable, deterministic and episodic. Each agent can see the full state $\mathbf{s}$, $f_P$ is deterministic and each episode $\tau = (\mathbf{s}_t, \mathbf{a}_t)_{0 \leq t \leq T}$ ends when the agent encounters a terminal state and the MDP resets.

In the fully cooperative case, the goal for each agent is to learn its optimal policy $P^*\left(a_t^i|\mathbf{s}_t, \boldsymbol{\theta}\right)$ that maximizes the total reward $R(\tau) = \sum_i R^i(\tau) = \sum_i \sum_t r^i(\mathbf{s}_t, \mathbf{a}_t)$:

$$\max_{\boldsymbol{\theta}} J(\boldsymbol{\theta}) = \max_{\boldsymbol{\theta}} \sum_{i \in \mathcal{I}} J^i(\boldsymbol{\theta}), \quad J^i(\boldsymbol{\theta}) = \mathbb{E}\left[R^i\left(\tau\right) \middle| \mathbf{a}_t \sim P(\mathbf{a}_t|\mathbf{s}_t; \boldsymbol{\theta})\right] \tag{1}$$

To optimize, we can apply gradient descent with policy gradient estimators $\hat{g}_{\boldsymbol{\theta}}$ (Williams (1992))

$$g_{\boldsymbol{\theta}} = \nabla_{\boldsymbol{\theta}} J\left(\boldsymbol{\theta}\right) = \mathbb{E}\left[\nabla_{\boldsymbol{\theta}} \log P(\mathbf{a}_t|\mathbf{s}_t; \boldsymbol{\theta})R(\tau)|\mathbf{a}_t, \mathbf{s}_t\right] \approx \frac{1}{M} \sum_{k=1}^{M} \sum_t \nabla_{\boldsymbol{\theta}} \log P(\mathbf{a}_t^k|\mathbf{s}_t^k; \boldsymbol{\theta})R(\tau^k), \tag{2}$$

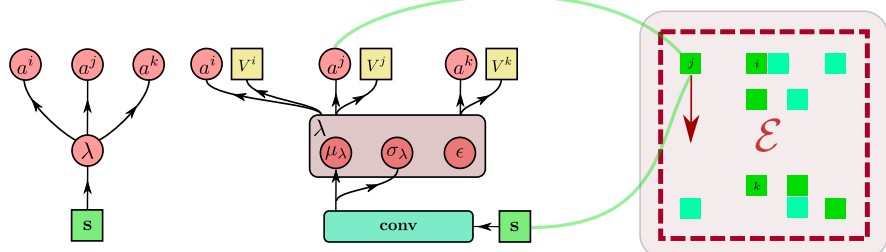

Figure 2: Structured latent variable model of the multi-agent policy (*actor*, left) and instance of the multi-agent actor-critic interacting in the environment $\mathcal{E}$ (right). The joint policy contains two stacked layers of stochastic latent variables (red), and deterministically receives states and computes actions (green). Global variables $\lambda$ are shared across agents. On the right, a neural network instance of the actor-critic uses the reparametrization trick and receives the environment state, samples actions from the policy for all agents and computes value functions $V^{\bullet}$.

where we sample $M$ rollouts $\tau^k$ by sampling actions from the policy that is being learned.

A central issue in reinforcement learning is the *exploration-exploitation* trade-off: how can agents sample rollouts and learn efficiently? In particular, when the state-action space is exponentially large, discovering good (coordinated) policies when each agent samples independently becomes combinatorially intractable as $N$ grows. Hence, exploration in large state-action spaces poses a significant challenge.

## 2.1 MACE: JOINT COORDINATION AND EXPLORATION

We now formulate our multi-agent objective (1) using a hierarchical policy class that enables structured exploration. Our approach, MACE ("Multi-Agent Coordinated Exploration"), builds upon two complementary approaches:

- Encode structured exploration by sampling actions that are *correlated* between agents. The correlation between actions encodes coordination.

- Use a variational approach to derive and optimize a lower bound on the objective (1).

**Hierarchical Latent Model.** To encode coordination between agents, we assume the individual policies have shared structure, encoded by a latent variable $\lambda_t \in \mathbb{R}^n$ for all $t$, where $n$ is the dimension of the latent space. This leads to a hierarchical policy model $P(\mathbf{a}_t, \lambda_t | \mathbf{s}_t)$, as shown in Figure 2. We first write the joint policy for a single time-step as:

$$P(\mathbf{a}_t|\mathbf{s}_t) = \int d\lambda_t P(\mathbf{a}_t, \lambda_t|\mathbf{s}_t) = \int d\lambda_t \prod_{i=1}^{N} P(a_t^i, \lambda_t|\mathbf{s}_t) = \int d\lambda_t \prod_{i=1}^{N} P(a_t^i|\lambda_t, \mathbf{s}_t)P(\lambda_t|\mathbf{s}_t), \quad (3)$$

where we introduced the conditional priors $P(\lambda_t|\mathbf{s}_t)$. The latent variables $\lambda_t$ introduce dependencies among the $\mathbf{a}_t$, hence this policy is more flexible compared to standard fully factorized policies (Ranganath et al. (2015)). Note that this approach supports centralized learning and decentralized execution, by sharing a random seed amongst agents to sample $\lambda_t$ and actions $\mathbf{a}_t$ during execution.

Computing the integral in the optimal policy (3) is hard, because the unknown distribution $P(a_t^i|\lambda_t, \mathbf{s}_t)$ can be highly complex. Hence, to make learning (3) tractable, we will use a variational approach.

**Hierarchical Variational Lower Bound.** We next derive a tractable learning algorithm using variational methods. Instead of directly optimizing (1), we cast it as a probabilistic inference problem, as in Levine & Koltun (2013); Vlassis et al. (2009), and instead optimize a lower bound. To do so, we assume that the total reward $R^i$ for each $i$ to be non-negative and bounded. Hence, we can view the total reward $R(\tau)$ as a *random variable*, whose *unnormalized* distribution is defined as $P(R|\tau) = R$. We can then rewrite (1) as a maximum likelihood problem:

$$\max_{\boldsymbol{\theta}} \mathbb{E}_{\pi(\mathbf{s}_t; \boldsymbol{\theta})}[R] = \max_{\boldsymbol{\theta}} \mathbb{E}_{\pi(\mathbf{s}_t; \boldsymbol{\theta})}[P(R|\tau)] \quad (4)$$

Hence, the RL objective is equivalent to a maximal likelihood problem:

$$\max_{\boldsymbol{\theta}} \mathbb{E}_{\pi(\mathbf{s}_t;\boldsymbol{\theta})}\left[P(R|\tau)\right] = \max_{\boldsymbol{\theta}} \int d\tau P(R|\tau)P(\tau;\boldsymbol{\theta}) \Leftrightarrow \max_{\boldsymbol{\theta}} \int d\tau \log P(R|\tau)P(\tau;\boldsymbol{\theta}). \quad (5)$$

In MACE we introduce a latent variable $\lambda_t$ in the probability of a rollout $\tau$, using (3):

$$P(\tau;\boldsymbol{\theta}) = P(\mathbf{s}_0) \int d\lambda_{0:T} \prod_{t=0}^{T} P(\mathbf{s}_{t+1}|\mathbf{s}_t, \mathbf{a}_t) P(\mathbf{a}_t, \lambda_t|\mathbf{s}_t; \boldsymbol{\theta}), \quad \int d\lambda_{0:T} \equiv \prod_{t=0}^{T} \int d\lambda_t, \quad (6)$$

where computing the policy distribution $P(\mathbf{a}_t, \lambda_t|\mathbf{s}_t; \boldsymbol{\theta})$ is intractable, which makes the maximization in Equation (5) hard. Hence, we derive a lower bound on the log-likelihood $\log P(R|\tau)P(\tau;\boldsymbol{\theta})$ in Equation (5), using a variational approach. Specifically, we use an approximate *factorized* variational distribution $Q_R$ that is weighted by the total reward $R$:

$$Q_R(\lambda_{0:T}|\tau; \boldsymbol{\phi}) = P(R|\tau)P(\mathbf{s}_0) \prod_{t=0}^{T} P(\mathbf{s}_{t+1}|\mathbf{s}_t, \mathbf{a}_t) Q(\lambda_t|\mathbf{s}_t; \boldsymbol{\phi}), \quad (7)$$

where $\boldsymbol{\phi}$ are the parameters for the variational distribution $Q_R$. Using Jensen's inequality (Hoffman et al., 2013) and (3) to factorize $P(\mathbf{a}_t, \lambda_t|\mathbf{s}_t; \boldsymbol{\theta})$, we can derive:

$$\log P(R|\tau)P(\tau;\boldsymbol{\theta}) \geq \underbrace{\int d\lambda_{0:T} Q_R(\lambda_{0:T}|\tau; \boldsymbol{\phi}) \sum_{t=0}^{T} \left( \log P(\mathbf{a}_t|\lambda_t, \mathbf{s}_t; \boldsymbol{\theta}) + \log \frac{P(\lambda_t|\mathbf{s}_t)}{Q(\lambda_t|\mathbf{s}_t, \boldsymbol{\phi})} \right)}_{\text{ELBO}(Q_R, \boldsymbol{\theta}, \boldsymbol{\phi})}, \quad (8)$$

where the right-hand side is called the evidence lower bound (ELBO), which we can maximize as a proxy for (4). For more details on the derivation, see the Appendix.

The standard choice for the prior $P(\lambda_t|\mathbf{s}_t)$ is to use maximum-entropy standard-normal priors: $P(\lambda_t|\mathbf{s}_t) = \mathcal{N}(\mathbf{0}, \mathbf{1})$. We can then optimize (8) using e.g. stochastic gradient ascent. Formally, the MACE policy gradient is:

$$g_{\boldsymbol{\theta}} \approx g_{\boldsymbol{\theta},Q} = \nabla_{\boldsymbol{\theta}}\text{ELBO}(Q_R, \boldsymbol{\theta}, \boldsymbol{\phi}) = \int d\tau d\lambda_{0:T} Q_R(\lambda_{0:T}|\tau; \boldsymbol{\phi}) \sum_{t'=0}^{T} \nabla_{\boldsymbol{\theta}} \log P(\mathbf{a}_{t'}|\lambda_{t'}, \mathbf{s}_{t'}; \boldsymbol{\theta}), \quad (9)$$

which is an approximation of the true policy gradient (2). This gradient can be estimated using sampled roll-outs $\tau^k$ of the policy $P^{\boldsymbol{\pi}}$:

$$g_{\boldsymbol{\theta},Q} \approx \hat{g}_{\boldsymbol{\theta},Q} = \frac{1}{M} \sum_{k=1}^{M} \sum_{t=0}^{T} \nabla_{\boldsymbol{\theta}} \log P\left(\mathbf{a}_t^k|\lambda_t^k, \mathbf{s}_t^k; \boldsymbol{\theta}\right) R(\tau^k). \quad (10)$$

During a rollout $\tau^k$, we sample $\lambda \sim Q$, observe rewards $R \sim P(R|\tau)$ and transitions $s_{t+1} \sim P(s_{t+1}|.)$, and use these to compute (10). We can similarly compute $g_{\boldsymbol{\phi},Q} = \nabla_{\boldsymbol{\phi}}\text{ELBO}(Q_R, \boldsymbol{\theta}, \boldsymbol{\phi})$, the gradient for the variational posterior $Q_R$.

**Actor-Critic and Bias-Variance.** Estimating policy gradients $g_{\boldsymbol{\theta}}$ using empirical rewards can suffer from high variance and instabilities. It is thus useful to consider more general objectives $F^i$:

$$J^i(\boldsymbol{\theta}) = \mathbb{E}\left[ F^i(\tau) \big| \mathbf{a}_t \sim P^{\boldsymbol{\pi}_t}(\mathbf{a}_t|\mathbf{s}_t; \boldsymbol{\theta}) \right], \quad (11)$$

such that the variance in $\hat{g}$ is reduced.[1] In practice, we find that using (10) with more general $F$, such as generalized advantages (Schulman et al. (2015)), performs quite well.

## 3 EXPERIMENTAL VALIDATION

### 3.1 MULTI-AGENT ENVIRONMENTS

To validate our approach, we created two grid-world games, depicted in Figure 2, inspired by the classic Predator-Prey and Stag-Hunt games (Shoham & Leyton-Brown (2008)). In both games, the

---

[1]Note that if the total reward $R$ is bounded, we can define $R$-weighted probabilities by shifting and normalizing $R$. In general, the derivation applies if $F$ is similarly bounded.

world is periodic and the initial positions of the hunters and prey are randomized. Also, we consider two instances for both games: either the prey are moving or fixed.

**Hare-Hunters.** Predator-Prey is a classic test environment for multi-agent learning, where 4 predators try to capture a prey by boxing it in. We consider a variation defined by the settings $(N, M, H, T)$: $N$ hunters and $M$ prey. Each prey can be captured by exactly $H$ hunters: to capture the prey, a hunter gets next to it, after which the hunter is frozen. Once a prey has had $H$ hunters next to it, it is frozen and cannot be captured by another hunter. The terminal rewards used are:

$$R^i = \begin{cases} 1, & \text{if } \textit{all} \text{ prey are captured } H \text{ times before the time limit } T \\ 0, & \text{otherwise} \end{cases} \qquad (12)$$

The challenge of the game is for the agents to inactivate all prey within a finite time $T$. Due to the time limit, the optimal strategy is for the agents to distribute targets efficiently, which can be challenging due to the combinatorially large number of possible hunter-to-prey assignments.

**Stag-Hunters.** The Stag-Hunt is another classic multi-agent game designed to study coordination. In this game, hunters have a choice: either they capture a hare for low reward, or, together with another hunter, capture a stag for a high reward. We extend this to the multi-agent $(N, M, H, T)$-setting: $N$ hunters hunt $M$ prey ($M/2$ stags and $M/2$ hares). Each stag has $H$ hit-points, while hares and hunters have 1 hit-point. Capturing is as in `Hare-Hunters`. The spatial domain is similar to the `Hare-Hunters` game and we also use a time limit $T$. The terminal reward is now defined as:

$$R^i = \begin{cases} 1, & \text{if } i \text{ captured a live stag that became inactive before the time limit } T \\ 0.1, & \text{if } i \text{ captured a live hare before the time limit } T \\ 0, & \text{otherwise} \end{cases} \qquad (13)$$

The challenge for the agents here is to discover that choosing to capture the same prey can yield substantially higher reward, but this requires coordinating with another hunter.

## 3.2 Neural Coordination Model

For experiments, we instantiated our multi-agent policy class (as in Figure 2) with deep neural networks. For simplicity, we only used *reactive policies* without memory, although it is straightforward to apply `MACE` to policies with memory (e.g. LSTMs). The model takes a joint state $\mathbf{s}_t$ as input and computes features $\phi(\mathbf{s})$ using a 2-layer convolutional neural network. To compute the latent variable $\lambda \in \mathbb{R}^d$, we use the reparametrization trick (Kingma & Welling, 2013) to learn the variational distribution (e.g. $Q(\lambda|\mathbf{s})$), sampling $\lambda$ via $\epsilon \sim \mathcal{N}(\mathbf{0}, \mathbf{1})$ and distribution parameters $\mu, \sigma$ (omitting $t$):

$$\mu(\mathbf{s}) = W_\mu \phi(\mathbf{s}) + b_\mu, \quad \log \sigma(\mathbf{s})^2 = W_\sigma \phi(\mathbf{s}) + b_\sigma, \quad \lambda = \mu(\mathbf{s}) + \sigma(\mathbf{s}) \odot \epsilon. \qquad (14)$$

Given $\lambda$, the model then computes the policies $P(a^i|\lambda, \mathbf{s})$ and value functions $V^i(\mathbf{s})$ as (omitting $t$):

$$P(a^i|\lambda, \mathbf{s}) = \texttt{softmax}\left(W_\pi^i[\lambda \ \phi(\mathbf{s})] + b_\pi^i\right), \quad V^i(\mathbf{s}) = W_V^i \phi(\mathbf{s}) + b_V^i, \qquad (15)$$

where $\texttt{softmax}(\boldsymbol{x}) = \exp \boldsymbol{x} / \sum_j \exp x^j$. In this way, the model can be trained end-to-end.

**Training.** We used A3C (Mnih et al. (2016)) with KL-controlled policy gradients (10), generalized advantage as $F$ (Schulman et al. (2015)). and policy-entropy regularization. For all experiments, we performed a hyper-parameter search and report the best 5 runs seen (see the Appendix for details).

**Baselines.** We compared `MACE` against two natural baselines:

- `Shared` (shared actor-critic): agents share a deterministic hidden layer, but maintain individual weights $\theta^i$ for their (stochastic) policy $P(a|\lambda, \mathbf{s}; \theta^i)$ and value function $V^i(\mathbf{s}; \boldsymbol{\theta})$. The key difference is that this model does not sample from the shared hidden layer.

- `Cloned` (actor-critic): a model where each agent uses an identical policy and value function with shared weights. There is shared information between the agents, and actions are sampled according to the agents' own policies.

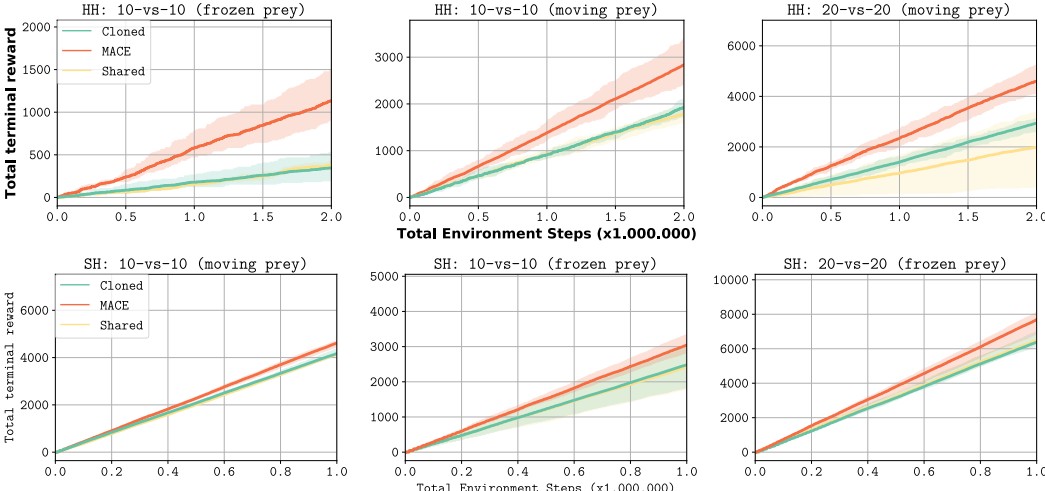

Figure 3: Train-time cumulative terminal reward for $N$ agents in $(N, M, 1, T)$ `Hare-Hunters` (upper, $T = 2000$) and `Stag-Hunters` (lower, $T = 1000$) on a $50 \times 50$ gridworld, for **10-vs-10** or **20-vs-20** agents; randomly moving or fixed preys. Average, minimal and maximal rewards for the best 5 runs for each model are shown. `MACE` accumulates increasingly higher rewards compared to the baselines, by 1) achieving higher terminal reward per episode and 2) finishing episodes faster (see Figure 4). For 10-10 `Stag-Hunters` with frozen prey, average reward per-episode is 4.64 (`Cloned`), 6.22 (`Shared`), 6.61 (`MACE`) after 1 million samples. For more, see the Appendix.

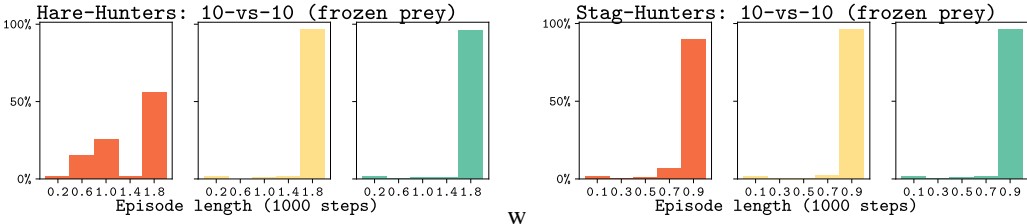

Figure 4: Train-time episode lengths during 1 million steps for **10-vs-10** `Hare-Hunters` (left) and `Stag-Hunters` (right), with fixed preys. `MACE` (orange) finishes an episode successfully before the time limit more often than the baselines (`Cloned` (blue) and `Shared` (yellow)).

## 4 QUANTITATIVE ANALYSIS

Above, we defined a variational approach to train hierarchical multi-agent policies using structured exploration. We now validate the efficacy of our approach by showing our method scales to environments with a large number of agents. We ran experiments for both `Hare-Hunters` and `Stag-Hunters` for $N = M = 10, 20$ in a spatial domain of $50 \times 50$ grid cells.

**Sample complexity.** In Table 1 we show the achieved rewards after a fixed number of training samples, and Figure 3 showcases the corresponding learning curves. We see that `MACE` achieves up to $10\times$ reward compared to the baselines. Figure 4 shows the corresponding distribution of training episode lengths. We see that `MACE` solves game instances more than $20\%$ faster than baselines in $50\%$ ($10\%$) of `Hare-Hunters` (`Stag-Hunters`) episodes. In particular, `MACE` learns to coordinate for higher reward more often: it achieves the highest average reward per-episode (e.g. for 10-10 `Stag-Hunters` with frozen prey, average rewards are 4.64 (`Cloned`), 6.22 (`Shared`), 6.61 (`MACE`)). Hence, `MACE` coordinates successfully more often to capture the stags. Together, these results show `MACE` enables more efficient learning.

**Using the ELBO.** A salient difference between (10) and (2) is the KL-regularization, which stems from the derivation of the ELBO. Since we use a more general objective $F$, c.f. (11), we also investigated the impact of using the KL-regularized policy gradient (10) versus the standard (2). To

| Preys are | Frozen | | Moving | | Frozen | | Moving | |
|---|---|---|---|---|---|---|---|---|
| Samples (x100k) | 5 | 10 | 5 | 10 | 5 | 10 | 5 | 10 |
| Hare-Hunters | **10-vs-10** | | | | **20-vs-20** | | | |
| Cloned | 85.0 | 178.3 | 465.0 | 912.5 | 20.0 | 70.0 | 706.7 | 1401.7 |
| Shared | 65.0 | 155.0 | 457.5 | 923.8 | 65.0 | 105.0 | 491.4 | 962.9 |
| MACE | **240.0** | **580.0** | **662.7** | **1381.8** | **200.0** | **393.3** | **1260.0** | **2344.0** |
| Stag-Hunters | **10-vs-10** | | | | **20-vs-20** | | | |
| Cloned | 1229.2 | 2482.9 | 2079.4 | 4171.2 | 3224.5 | 6219.9 | 5934.5 | 11429.2 |
| Shared | 1214.5 | 2423.7 | 2005.7 | 4144.3 | 3150.7 | 6379.8 | 6344.4 | 12196.8 |
| MACE | **1515.2** | **3047.3** | **2275.7** | **4610.7** | **3799.3** | **7158.1** | **6880.7** | **13358.6** |

Table 1: Total terminal reward (averaged over 5 best runs) for $N$ agents for set # of training samples.

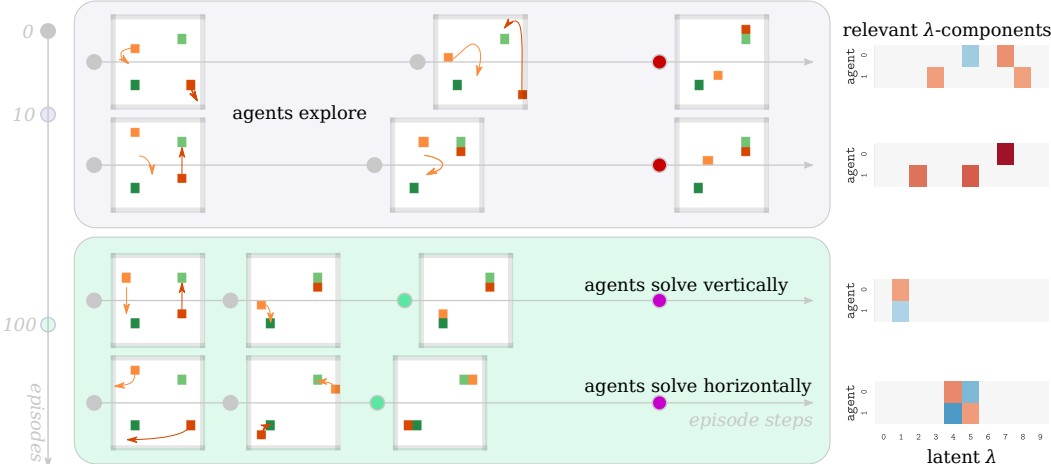

Figure 5: Predators (red) and prey (green) during training for 2v2 Hare-Hunters for 100 episodes. Arrows show where agents move to in the next frame. Top: at the start, predators explore via $\lambda$, but do not succeed before the time limit $T$ (red dot). Bottom: after convergence agents succeed consistently (green dot) before the time limit (purple dot) and $\lambda$ encodes the two strategies from Figure 1. Highlighted $\lambda$-components correlate with rollout under a 2-sided $t$-test at $\alpha = 0.1$ significance.

this end, we ran several instances of the above experiments both with and without KL-regularization. We found that without KL-regularization, training is unstable and prone to mode collapse: the variance $\sigma$ of the variational distribution can go to 0. This reflects in essentially 0 achieved reward: the model does not solve the game for any reasonable hyperparameter settings.

**Impact of dynamics and $T$.** Inspecting training performance, we see the relative difficulty of capturing moving or randomly moving prey. Capturing moving prey is easier to learn than capturing fixed preys, as comparing rewards in Table 1 shows. This shows a feature of the game dynamics: the expected distance between a hunter and an uncaptured prey are lower when the preys are randomly moving, resulting in an easier game. Comparing Hare-Hunters and Stag-Hunters, we also see the impact of the time limit $T$. Since we use terminal rewards only, as $T$ gets larger, the reward becomes very sparse and models need more samples to discover good policies.

## 5   MODEL INSPECTION

Beyond training benefits, we now demonstrate empirical evidence that suggest efficacy and meaningfulness of our approach to structured exploration. We start by inspecting the behavior of the latent variable $\lambda$ for a simple $N = M = 2$ Hare-Hunters game, which enables semantic inspection of the learned policies, as in Figure 5. We make a number of observations. First, $\lambda$ is relevant: many components are statistically significantly correlated with the agents' actions. This suggests the model

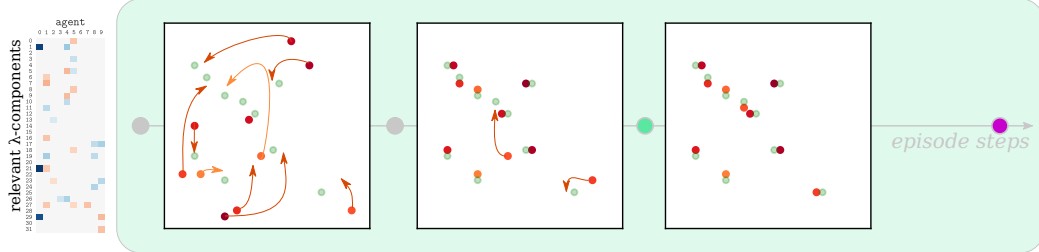

Figure 6: Visualization of `MACE` for a $(10, 10, 1, 1000)$ `Hare-Hunters` game in a $30 \times 30$ world. Left: components of the latent code $\lambda$ that significantly correlate with sampled actions (computed as in Figure 5). Right: Three episode snapshots: at the start, middle and end. Red: predators; green: prey. Arrows indicate where agents move to in the next snapshot. The hunters solve the game (green dot) before the time limit $T = 1000$ (purple dot), by distributing targets amongst themselves.

does indeed use the latent $\lambda$: it (partly) controls the coordination between agents.[2] Second, the latent $\lambda$ shows strong correlation during all phases of training. This suggests that the model indeed is performing a form of structured exploration. Third, the components of $\lambda$ are correlated with semantic meaningful behavior. We show a salient example in the bottom 2 rows in Figure 5: the correlated components of $\lambda$ are disjoint and each component correlates with both agents. The executed policies are exactly the two equivalent ways to assign 2 hunters to 2 preys, as illustrated in Figure 1.

**Coordination with a large $N$.** In the large $N = M = 10$ case, the dynamics of the agent collective are a generalization of the $N = M = 2$ case. There are now redundancies in multi-agent hunter-prey assignments that are analogous to the $N = M = 2$ case that are prohibitively complex to analyze due to combinatorial complexity. However our experiments strongly suggest (see e.g. Figure 6) the latent code is again correlated with the agents' behavior during all phases of training, showing that $\lambda$ induces meaningful multi-agent coordination.

## 6 RELATED WORK

**Deep Structured Inference.** Recent works have focused on learning structured representations using expressive distributions, which enable more powerful probabilistic inference. For instance, Johnson et al. (2016) has proposed combining neural networks with graphical models, while Ranganath et al. (2015) learn hierarchical latent distributions. Our work builds upon these approaches to learn structured policies in the reinforcement learning setting. In the multi-agent setting, the RL problem has also been considered as an inference problem in e.g. (Liu et al., 2015; Wu et al., 2013; Liu et al., 2016).

**Variational methods in RL.** Neumann (2011); Furmston & Barber (2010) discuss variational approaches for RL problems, but did not consider end-to-end trainable models. Levine & Koltun (2013) used variational methods for guided policy search. Houthooft et al. (2016) learned exploration policies via information gain using variational methods. However, these only consider 1 agent.

**Coordination in RL.** Multi-agent coordination has been studied in the RL community (e.g. Guestrin et al. (2002); Kapetanakis & Kudenko (2002); Chalkiadakis & Boutilier (2003)), for instance, as a method to reduce the instability of multiple agents learning simultaneously using RL. The benefit of coordination was already demonstrated in simple multi-agent settings in e.g. Tan (1993). The shared latent variable $\lambda$ of our structured policy can also be interpreted as a learned *correlation device* (see Bernstein (2005) for an example in the decentralized setting), which can be used to e.g. break ties between alternatives or induce coordination between agents. More generally, they can be used to achieve correlated equilibria (Greenwald & Hall, 2003), a more general solution concept than Nash equilibria. However, previous methods learned hand-crafted models and do not scale well to complex state spaces and many agents. In contrast, our method learns coordination end-to-end via on-policy methods, *learns* the multi-agent exploration policy and scales well to many agents via its simple hierarchical structure.

---

[2]This is parallel to the discussion in Chen et al. (2016), which investigates the effectiveness of latent codes $\lambda$.

**Communication Models.** Recently, end-to-end learning of communication models has been studied for shared broadcast channels (Sukhbaatar et al. (2016)), sequential communication (Peng et al. (2017)), heuristic multi-agent exploration Usunier et al. (2016) and bit-channels (Foerster et al. (2016)). These works show that non-trivial communication protocols can be learned through backpropagation or heuristic stabilization methods, but often do not scale well to a large number of agents. Our hierarchical approach is complementary, learns via variational methods, and can scale to large $N$.

**Multi-task learning.** Hierarchical models have been studied for multi-task learning, e.g. Daume III (2014) learns latent hierarchies via EM in a supervised learning setting. Instead, we study flexible end-to-end trainable latent hierarchies in the reinforcement learning setting.

## 7 DISCUSSION

In a sense, we studied the simplest setting that can benefit from structured exploration, in order to isolate the contribution of our work. Our hierarchical model and variational approach are a simple way to implement multi-agent coordination, and easily combine with existing actor-critic methods. Moving forward, there are many ways to expand on our work. Firstly, for complex (partial-information) environments, instead of using reactive policies with simple priors $P \sim \mathcal{N}(0, 1)$, memoryfull policies with flexible priors (Chen et al., 2016) may be needed. Secondly, our approach is complementary to richer forms of communication between agents. Our hierarchical structure can be interpreted as a broadcast channel, where agents are passive receivers of the message $\lambda$. Richer communication protocols could be encoded by policies with more complex inter-agent structure. It would be interesting to investigate how to learn these richer structures.

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

# 8 SUPPLEMENTARY MATERIAL

## 8.1 HIERARCHICAL VARIATIONAL LOWER BOUND.

We show details on how to derive a tractable learning method to the multi-agent reinforcement learning problem with a centralized controller:

$$R = \sum_i R^i, \quad \max_{\boldsymbol{\theta}} \mathbb{E}_{\pi(\mathbf{s}_t; \boldsymbol{\theta})}[R]. \tag{16}$$

Instead of directly optimizing (16), we cast it as a probabilistic inference problem, as in Levine & Koltun (2013); Vlassis et al. (2009), and optimize a lower bound.

To do so, we assume that the total reward $R^i$ for each $i$ to be non-negative and bounded. Hence, we can view the total reward $R(\tau)$ as a *random variable*, whose *unnormalized* distribution is defined as

$$P(R|\tau) = R. \tag{17}$$

We can then rewrite (16) as a maximum likelihood problem:

$$\max_{\boldsymbol{\theta}} \mathbb{E}_{\pi(\mathbf{s}_t; \boldsymbol{\theta})}[R] = \max_{\boldsymbol{\theta}} \mathbb{E}_{\pi(\mathbf{s}_t; \boldsymbol{\theta})}[P(R|\tau)] \tag{18}$$

Hence, the RL objective is equivalent to a maximal likelihood problem:

$$\max_{\boldsymbol{\theta}} \mathbb{E}_{\pi(\mathbf{s}_t; \boldsymbol{\theta})}[P(R|\tau)] = \max_{\boldsymbol{\theta}} \int d\tau \underbrace{P(R|\tau)P(\tau; \boldsymbol{\theta})}_{P(R, \tau; \boldsymbol{\theta})} \Leftrightarrow \max_{\boldsymbol{\theta}} \int d\tau \log P(R|\tau)P(\tau; \boldsymbol{\theta}), \tag{19}$$

where the probability of a rollout $\tau$ features a marginalization over the latent variables $\lambda_t$:

$$P(\tau; \boldsymbol{\theta}) = P(\mathbf{s}_0) \prod_{t=0}^{T} P(\mathbf{s}_{t+1}|\mathbf{s}_t, \mathbf{a}_t) P(\mathbf{a}_t|\mathbf{s}_t; \boldsymbol{\theta})$$

$$= P(\mathbf{s}_0) \int d\lambda_{0:T} \prod_{t=0}^{T} P(\mathbf{s}_{t+1}|\mathbf{s}_t, \mathbf{a}_t) P(\mathbf{a}_t, \lambda_t|\mathbf{s}_t; \boldsymbol{\theta}), \tag{20}$$

$$\int d\lambda_{0:T} \equiv \prod_{t=0}^{T} \int d\lambda_t \tag{21}$$

Here, we used the hierarchical decomposition for the policy:

$$P(\mathbf{a}_t|\mathbf{s}_t; \boldsymbol{\theta}) = \int d\lambda_t P(\mathbf{a}_t, \lambda_t|\mathbf{s}_t; \boldsymbol{\theta}) \tag{22}$$

$$= \int d\lambda_t \prod_{i=1}^{N} P(a_t^i, \lambda_t|\mathbf{s}_t; \boldsymbol{\theta}) \tag{23}$$

$$= \int d\lambda_t \prod_{i=1}^{N} P(a_t^i|\lambda_t, \mathbf{s}_t; \boldsymbol{\theta}) P(\lambda_t|\mathbf{s}_t), \tag{24}$$

This policy distribution is intractable to learn exactly, as it involves margalization over $\lambda_t$ and an unknown flexible distribution $P(a_t^i|\lambda_t, \mathbf{s}_t)$. Hence the maximization in Equation (19) is hard. Hence, we follow the variational approach and get a lower bound on the log-likelihood $\log P(R, \tau; \boldsymbol{\theta})$ in Equation (19). For this, we use an *approximate* variational distribution $Q_R(\lambda_{0:T}|\tau; \phi)$ and Jensen's inequality (Hoffman et al., 2013):

$$\log P(R, \tau; \boldsymbol{\theta}) \geq \int d\lambda_{0:T} Q_R(\lambda_{0:T}|\tau; \phi) \log \left( \frac{P(R, \tau; \boldsymbol{\theta})}{Q_R(\lambda_{0:T}|\tau; \phi)} \right) \tag{25}$$

$$= \int d\lambda_{0:T} Q_R(\lambda_{0:T}|\tau; \phi) \log \left( \frac{P(R|\tau)P(\mathbf{s}_0) \prod_{t=0}^{T} P(\mathbf{s}_{t+1}|\mathbf{s}_t, \mathbf{a}_t) P(\mathbf{a}_t, \lambda_t|\mathbf{s}_t; \boldsymbol{\theta})}{Q_R(\lambda_{0:T}|\tau; \phi)} \right), \tag{26}$$

where in the last line we used (20). By inspecting the quotient in (26), we see that the optimal $Q_R$ is a factorized distribution weighted by the total reward $R$:

$$Q_R(\lambda_{0:T}|\tau;\phi) = P(R|\tau)P(\mathbf{s}_0)\prod_{t=0}^{T}P(\mathbf{s}_{t+1}|\mathbf{s}_t,\mathbf{a}_t)Q(\lambda_t|\mathbf{s}_t;\phi). \tag{27}$$

We see that (26) simplifies to:

$$\int d\lambda_{0:T}Q_R(\lambda_{0:T}|\tau;\phi)\log\left(\frac{P(R|\tau)P(\mathbf{s}_0)\prod_{t=0}^{T}P(\mathbf{s}_{t+1}|\mathbf{s}_t,\mathbf{a}_t)P(\mathbf{a}_t,\lambda_t|\mathbf{s}_t;\boldsymbol{\theta})}{P(R|\tau)P(\mathbf{s}_0)\prod_{t=0}^{T}P(\mathbf{s}_{t+1}|\mathbf{s}_t,\mathbf{a}_t)Q(\lambda_t|\mathbf{s}_t;\phi)}\right) \tag{28}$$

$$=\int d\lambda_{0:T}Q_R(\lambda_{0:T}|\tau;\phi)\log\left(\prod_{t=0}^{T}\frac{P(\mathbf{a}_t,\lambda_t|\mathbf{s}_t;\boldsymbol{\theta})}{Q(\lambda_t|\mathbf{s}_t,\phi)}\right) \tag{29}$$

$$=\int d\lambda_{0:T}Q_R(\lambda_{0:T}|\tau;\phi)\sum_{t=0}^{T}\left(\log\frac{P(\mathbf{a}_t,\lambda_t|\mathbf{s}_t;\boldsymbol{\theta})}{Q(\lambda_t|\mathbf{s}_t,\phi)}\right) \tag{30}$$

$$=\int d\lambda_{0:T}Q_R(\lambda_{0:T}|\tau;\phi)\sum_{t=0}^{T}\left(\log\frac{P(\mathbf{a}_t|\lambda_t,\mathbf{s}_t;\boldsymbol{\theta})P(\lambda_t|\mathbf{s}_t)}{Q(\lambda_t|\mathbf{s}_t,\phi)}\right) \tag{31}$$

$$=\underbrace{\int d\lambda_{0:T}Q_R(\lambda_{0:T}|\tau;\phi)\sum_{t=0}^{T}\left(\log P(\mathbf{a}_t|\lambda_t,\mathbf{s}_t;\boldsymbol{\theta})+\log\frac{P(\lambda_t|\mathbf{s}_t)}{Q(\lambda_t|\mathbf{s}_t,\phi)}\right)}_{\texttt{ELBO}(Q_R,\boldsymbol{\theta},\phi)}. \tag{32}$$

The right-hand side in Equation (32) is called the evidence lower bound (`ELBO`), which we can maximize as a proxy for (16). The standard choice is to use maximum-entropy standard-normal priors: $P(\lambda_t|\mathbf{s}_t)=\mathcal{N}(\mathbf{0},\mathbf{1})$. We can then optimize (32) using e.g. stochastic gradient ascent.

## 8.2 TRAINING

**Training method.** We used the A3C method with 5-20 threads for all our experiments. Each thread performed SGD with (**??**). The loss for the value function at each state $\mathbf{s}_t$ is the standard $L_2$-loss between the observed total rewards for each agent $i$ and its value estimate:

$$\alpha\sum_t\sum_i\left(V^i(\mathbf{s}_t)-R^i(\mathbf{s}_t,\mathbf{a}_t)\right)^2. \tag{33}$$

In addition, in line with other work using actor-critic methods, we found that adding a small entropy regularization on the policy can sometimes positively influence performance, but this does not seem to be always required for our testbeds. The entropy regularization is:

$$H(P)=-\beta\sum_t\sum_{\mathbf{a}_t}P(\mathbf{a}_t|\mathbf{s}_t;\boldsymbol{\theta})\log P(\mathbf{a}_t|\mathbf{s}_t;\boldsymbol{\theta}). \tag{34}$$

A3C additionally defines training minibatches in terms of a fixed number of environment steps $L$: a smaller $L$ gives faster training with higher variance and a higher $L$ vice versa.

**Hyperparameter search.** For experiments, we performed a random search over hyperparameters:

| Hyperparameter | | Search range |
|---|---|---|
| Learning-rate | $\eta$ | $[10^{-6}, 10^{-1}]$ |
| Discount | $\gamma$ | $[0.9, 0.99]$ |
| GAE discount | $\tau$ | $[0.9, 0.99]$ |
| Value loss | $\alpha$ | $[0.1, 0.5]$ |
| Policy entropy | $\beta$ | $[0, 0.01]$ |

Table 2: Search range of hyper-parameters.

## 8.3 RESULTS

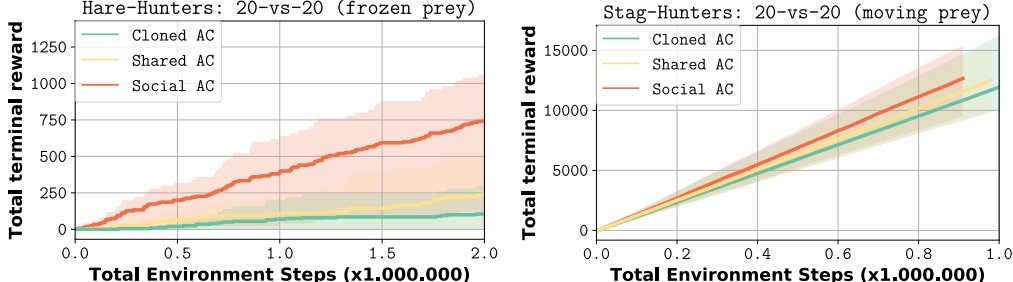

Figure 7: Train-time cumulative terminal reward in $(N, M, 1, 1000)$ Hare-Hunters (upper) and Stag-Hunters (lower) on a $50 \times 50$ gridworld, for **10-vs-10** or **20-vs-20** agents; randomly moving or fixed preys. Average, minimal and maximal rewards for the best 5 runs for each model are shown. MACE accumulates increasingly higher rewards compared to the baselines.

