# OpenReview forum: "Structured Exploration via Hierarchical Variational Policy Networks"
_ICLR.cc/2018/Conference — Reject_

### Official Review · AnonReviewer1 · 2017-11-23
**Seems interesting, but with significant weaknesses**

**Rating:** 4
**Confidence:** 5

**Review:**

This paper proposes an approach to improve exploration in multiagent reinforcement learning by allowing the policies of the individual agents to be conditioned on an external coordination signal \lambda. In order to find such parametrized policies, the approach combines deep RL with a variational inference approach (ELBO optimization). The paper presents an empirical evaluation, which seems encouraging, but that is also somewhat difficult to interpret given the lack of comparison to other state-of-the-art methods.

Overall, the paper seems interesting, but (in addition to the not completely convincing empirical evaluation), it has two main weaknesses: lack of clarity and grounding in related literature.

=Issues with clarity=

-"This problem has two equivalent solutions".
This is not so clear; depending on the movement of the preys it might well be that the optimal solution will switch to the other prey in certain cases?

-It is not clear what is really meant with the term "structured exploration". It just seems to mean 'improved'?

-It is not clear that the improvements are due to exploration; my feeling is that is is due to improved statistical strength on a more abstract state feature (which is learned), not unlike:
Geramifard, Alborz, et al. "Online discovery of feature dependencies." Proceedings of the 28th International Conference on Machine Learning (ICML-11). 2011.
However, there is no clear indication that there is an improved exploration policy.

-The problem setting is not quite clear:
The paper first introduces "multi-agent RL", which seems to correspond to a "stochastic game" (also "Markov game"), but then moves on to restrict to the "fully cooperative setting" (which would make it a "Multiagent MDP", Boutilier '96).

It subsequently says it deals only with deterministic problems (which would reduce the problem further to a learning version of a multiagent classical planning problem), but in the experiments do consider stochastically moving preys.

-The paper says the problem is fully observable, but fails to make explicit if this is *individually* fully observable, or jointly. I am assuming the former, but is it not clear how the agents observe this full state in the experimental evaluation.

This is actually a crucial confusion, as it completely changes the interpretation of what the approach does: in the individually observable case, the approach is adding a redundant source of information which is more abstract and thus seems to facilitate faster learning. In the latter case, where agents would have individual observations, it is actually providing the agents with more information.

As such, I would really encourage the authors to better define the task they are considering. E.g., by building on the taxonomies of problems that researchers have developed in the community focusing on decentralized POMDPs, such as:
Goldman, Claudia V., and Shlomo Zilberstein. "Decentralized control of cooperative systems: Categorization and complexity analysis." (2004).

-"Compared to the single-agent RL setting, multi-agent RL poses unique difficulties. A central issue
is the exploration-exploitation trade-off"
That now in particular happens to be a central issue in single agent RL too.

-"Finding the true posteriors P (λ t |s t ) ∝ P (s t |λ t )P (λ t ) is intractable in general"
The paper did not explain how this inference task is required to solve the RL problem.

-In general, I found the technical description impossible to follow, even after carefully looking at the appending. For instance, (also) there the term P (λ |s ) is suddenly introduced without explaining what the term exactly is? Why is the term P(a|λ) not popping up here? That also needs to be optimized, right? I suppose \phi is the parameter vector of the variational approximation, but it is never really stated. The various shorthand notations introduced for clarity do not help at all, but only make the formulas very cryptic.

-The main text is not readable since definitions, e.g., L(Q_r,\tehta,\phi), that are in the appendix are now missing.

-It is not clear to me how the second term of (10) is now estimated?

-"Shared (shared actor-critic): agents share a deterministic hidden layer,"
What kind of layer is this exactly? How does it relate to \lambda ?

-"The key difference is that this model does not sample from the shared hidden layer"
Why would sampling help? Given that we are dealing with a fully observable multiagent MDP, there is no inherent need to randomize at all? (there should be a optimal deterministic joint policy?)

-"There is shared information between the agents"
What information is referred to exactly?
Also: It is not quite clear if for these domains cloned would be better than completely independent learners (without shared weights)?

-I can't seem to find anywhere what is the actual shape (or type? I am assuming a vector of reals) of the used \lambda.

-in figure 5, rhs, what is being shown exactly? What do the colors mean? Why does there seem to be a \lambda *per* agent now?



=Related work=

I think the paper could/should be hugely improved in this respect.

The idea of casting MARL as inference has also been considered by:

Learning for Decentralized Control of Multiagent Systems in Large, Partially-Observable Stochastic Environments.
M Liu, C Amato, EP Anesta, JD Griffith, JP How - AAAI, 2016

Stick-breaking policy learning in Dec-POMDPs
M Liu, C Amato, X Liao, L Carin, JP How
International Joint Conference on Artificial Intelligence (IJCAI) 2015

Wu, F.; Zilberstein, S.; and Jennings, N. R. 2013. Monte-carlo
expectation maximization for decentralized POMDPs. In Proc.
of the 23rd Int’l Joint Conf. on Artificial Intelligence (IJCAI-
13).

I do not think that these explicitly make use of a mechanism to coordinate the policies, since they address to true Dec-POMDP setting where each agent only gets its own observations, but in the Dec-POMDP literature, there also is the notion of 'correlation device', which is an additional controller (say corresponding to a dummy agent), which of which the states can be observed by other agents and used to condition their actions on:

Bernstein DS, Hansen EA, Zilberstein S. Bounded policy iteration for decentralized POMDPs. InProceedings of the nineteenth international joint conference on artificial intelligence (IJCAI) 2005 Jun 6 (pp. 52-57).

(and clearly this could be directly included in the aforementioned learning approaches).


This notion of a correlation device also highlights to potential relation to methods to learn/compute correlated equilibria. E.g.,:

Greenwald A, Hall K, Serrano R. Correlated Q-learning. In ICML 2003 Aug 21 (Vol. 3, pp. 242-249).


A different connection between MARL and inference can be found in:

Zhang, Xinhua and Aberdeen, Douglas and Vishwanathan, S. V. N., "Conditional Random Fields for Multi-agent Reinforcement Learning", in (New York, NY, USA: ACM, 2007), pp. 1143--1150.


The idea of doing something hierarchical of course makes sense, but also here there are a number of related papers:

-putting "hierarchical multiagent" in google scholar finds works by Ghavamzadeh et al., Saira & Mahadevan, etc.

-Victor Lesser has pursued coordination for better exploration with a number of students.

I suppose that Guestrin et al.'s classical paper:
Guestrin, Carlos, Michail Lagoudakis, and Ronald Parr. "Coordinated reinforcement learning." ICML. Vol. 2. 2002.
would deserve a citation, and the MARL field is moving ahead fast, an explanation of the differences with COMA:
Counterfactual Multi-Agent Policy Gradients
J Foerster, G Farquhar, T Afouras, N Nardelli, S Whiteson
AAAI 2018
is probably also warranted.

---

### Official Review · AnonReviewer3 · 2017-11-27
**Paper presents interesting and potentially novel method.**

**Rating:** 7
**Confidence:** 3

**Review:**

The paper proposes a method to coordinate agent behaviour  by using policies that have a shared latent structure. The authors derive a variational policy optimisation method to optimise the coordinated policies. The approach is investigated empirically on 2 predator prey type games.

The method presented in the paper seems quite novel. The authors present a derivation of their variational, hierarchical  update. Not all steps in this derivation are equally well explained, especially the introduction of the variational posterior could be more detailed. The appendix also offers very little extra information compared to the main text, most paragraphs concerning the derivations are identical. The comparison to existing approaches using variational inference is quite brief. It would be nice to have a more detailed explanation of the novel steps in this approach.

 It also seems that by assuming a shared model, shared global state and a fully cooperative problem, the authors remove many of the complexities of a multi-agent system. This also brings the derivations closer to the single agent case.

A related potential criticism is the feasibility of using this approach in a multi-agent system. The authors are essentially creating a (partially) centralised learner. The cooperative rewards and shared structure assumptions structures mentioned above seem limiting in a multi-agent system. Even giving each agent local state observations is known to potentially create coordination problems. The predator prey games where agents with agents physically distributed over the environment are probably not the best motivational examples.

Other remarks:

Empirical result show a clear advantage for this method over the baselines. The evaluation domains are relatively simple, but it was nice to see that the authors also make an attempt to investigate the qualitative behaviour of their method.

The overview of related work was relatively brief and focused mostly on recent deep MARL approaches. There is a very large body on coordination in multi-agent RL. It would be nice to situate the research somewhat better within this field (or at least refer to an overview such as Busoniu et al, 2010).

It seems like a completely factorised approach (i.e. independent agents) would make a nice baseline for the experiments, in addition to the shared architecture approaches.

---

### Official Review · AnonReviewer2 · 2017-11-28
**Interesting method, missing related work and baselines, very limited experiments. Unclear if this is really a 'multi-agent' paper**

**Rating:** 5
**Confidence:** 3

**Review:**

This paper suggests an interesting algorithmic innovation, consisting of hierarchical latent variables for coordinated exploration in multi-agent settings.

Main concern: This work heavily relies on the multi-agent aspect for novelty :
"Houthooft et al. (2016) learned exploration policies via information gain using variational methods. However, these only consider 1 agent".  However, in the current form of the paper this is a questionable claim. As the problems investigated combine fully observable states, purely cooperative payouts and global latent variables, they reduce to single agent problems with a large action space. Effectively the 'different agents' are nothing but a parameterized action space of a central controller.
Using hierarchical latent variables for large action spaces is like a good idea, but placing the work into multi-agent seems like a red herring.

Given that this is a centralized controller, it would be really helpful to compare quantitatively to other approaches for structured exploration, eg [3] and [4].

Detailed comments:
-"we restrict to fully cooperative MDPs that are fully observable, deterministic and episodic." Since the rewards are also positive, a very relevant baseline (from a MARL point of view) is distributed Q-learning [1].
-Figure 3: Showing total cumulative terminal rewards is difficult to interpret. I would be interested in seeing standard 'training curves' which show the average return per episode after a given amount of training episodes. Currently it is difficult to judge whether training has converged on not.
-Related work is missing a lot of relevant research. Apart from references below, please see [2] for a relevant, if dated, overview.
-"Table 1: Total terminal reward (averaged over 5 best runs)" - how does the mean and median compare across methods for all runs rather than just the top 5?


References:
[1] Lauer, M., Riedmiller, M.: An algorithm for distributed reinforcement learning in cooperative
multi-agent systems. In: Proceedings 17th International Conference on Machine Learning
(ICML-00), pp. 535–542. Stanford University, US (2000)
[2] L. Bus¸oniu, R. Babuska, and B. De Schutter: Multi-Agent Reinforcement Learning: An Overview
[3] Nicolas Usunier, Gabriel Synnaeve, Zeming Lin, Soumith Chintala: Episodic Exploration for Deep Deterministic Policies: An Application to StarCraft Micromanagement Tasks
[4] Matthias Plappert, Rein Houthooft, Prafulla Dhariwal, Szymon Sidor, Richard Y. Chen, Xi Chen, Tamim Asfour, Pieter Abbeel, Marcin Andrychowicz: Parameter Space Noise for Exploration

---

### Public Comment · ~Zhanzhan_Zhao1 · 2017-12-07
**asking for the code**

I am a master student at Umich. We are now taking part in ICLR 2018 Reproducibility Challenge. So I wonder if I can get your code for the reproduce of the results in your paper. It is encouraged to get the code in this challenge and do some further investigations. So it will be great if you could offer us the code. My Email address is zhanzhao@umich.edu.  Thank you so much!

---

> ### Public Comment · (anonymous) · 2017-12-11
> **zhanzhao@umich.edu**
>
> zhanzhao@umich.edu

---

> ### Author Response · Authors · 2018-01-02
> **re**
>
> Thank you for your interest! We're working on cleaning the code and making it easy to use. We will let you know as soon as possible when we have a version that can be shared.

---

### Public Comment · (anonymous) · 2017-12-23
**Unclear in many ways**

We think this paper is not a clear paper. First, the definition of symbols in the paper is not clear. Many symbols used in the paper is not defined. Second, the decomposition of distribution of latent variables in the paper is not clear and looks less reasonable. Third, it is not clear how the agents observe this full states of the environment, etc. The paper states that the architecture of their approach helps the coordination of agents. But we think it is nothing but a centralized controller and we do not know the advantage of it over centralized controller method. To sum up, we think this paper is not in a high quality.

---

> ### Author Response · Authors · 2018-01-02
> **Clarifications**
>
> Thanks for your interest in our paper!
>
> "We think this paper is not a clear paper. First, the definition of symbols in the paper is not clear. Many symbols used in the paper is not defined."
> - We've uploaded a paper revision that clarifies the writing and definitions. Please let us know if there are further details that are unclear, so we can clarify further if needed.
>
> "Second, the decomposition of distribution of latent variables in the paper is not clear and looks less reasonable."
> - We model the policy distribution P(a_t|s_t) = \int dlambda P(a_t, lambda_t | s_t) using a latent variable lambda_t for each time-step t. The distribution of the latent variables lambda_t is learned by Q(lambda_t | s_t), which gives a principled variational lower bound on the log-likelihood of the policy distribution (and hence the expected reward). This is (one of) the simplest latent structure that one can assume, and that can capture a wide range of complex policy distributions P(a|s), given powerful neural network models for Q(lambda_t | s_t) and the "decoder" P(a|lambda, s).
>
> "Third, it is not clear how the agents observe this full states of the environment, etc. The paper states that the architecture of their approach helps the coordination of agents. But we think it is nothing but a centralized controller and we do not know the advantage of it over centralized controller method."
>
> - The centralized controller observes the positions of all agents (predators and prey). The coordination of the predators' actions is correlated through the latent variable lambda.
>
> It is unclear to us what is meant by "centralized controller method". We invite the commenter to clarify this.

---

### Public Comment · (anonymous) · 2017-12-23
**Beautiful Paper**

Really enjoyed reading the paper, one of better papers at ICLR in our opinion. It seems to us that you should be able to extend this to partially observable scenarios as well, and not just fully observable. Could you comment where things break down when assuming partially observability?

---

> ### Author Response · Authors · 2018-01-02
> **Partial observability**
>
> Thank you for your interest and response!
>
> We think the partial observable setting is very interesting to study wrt modeling policies as deep graphical models. In the partial observable case, one can choose a multi-agent learning problem definition to operate under. This has implications for how the joint / individual policies factorize / are shared. However, our method of modeling the hidden structure of the joint / individual policies can largely be generalized to these settings and are complementary to other works that operate under partial observability.
>
> 1. For instance, one can provide only partial observations to the individual agent's policies, but give the model for lambda access to the full state. Then, the model for lambda can be interpreted as a sort of "correlation device". In this case, our variational approach and policy factorization can still be applied.
>
> 2. Agent policies can be fully decoupled, where each agent now maintains their own coordination model (e.g. Q(lambda | observation_i). In this case, each agent can e.g. predict the observations / actions of other agents as well (see e.g. "Multi-Agent Actor-Critic for Mixed Cooperative-Competitive Environments", Lowe et al.). This could give the individual agents a way to learn an implied coordination model from their imputations of other agents' state. Here, the latent variables could compactly encode distributions that are efficient for state/action imputation for many agents simultaneously.
>
> Note that in our paper, the policy support centralized learning and decentralized execution, where the agents only have to share a random seed to sample lambdas during execution. Similarly, in the fully decoupled case, agents could still agree on a random seed during training and execution.

---

### Author Response · Authors · 2018-01-05
**Revision**

We'd like to notify reviewers that based on the reviews, we have uploaded a revision of our paper that addresses their comments and suggestions. Revised sections:

1. Title
2. Abstract
3. Introduction
4. Theory
5. Related work
6. Appendix

---

### Decision · Program_Chairs · 2018-01-29
**ICLR 2018 Conference Acceptance Decision**

**Decision:**

Reject

**Comment:**

The reviewers feel there are two issues that make this paper fall short of acceptance: first, the
lack of a clear emphasis and focus (evidenced by the significant revisions) and second, a lack of
comparison to similar, existing methods for multi-agent reinforcement learning.